# Cost profiles of cancer patients at the end of life: Estimates from the EPICOST-study

**Stefano Guzzinati**[1], **Alessandra Andreotti**[1], **Tania Lopez**[2], **Silvia Gori**[3]*, **Alberto Gagliani**[4], **Sandra Mallone**[2], **Daniela Pierannunzio**[2], **Andrea Tavilla**[2], **Alessandra Buja**[4], **Manuel Zorzi**[1], **Silvia Francisci**[2]

1 Veneto Cancer Registry, Epidemiological Department, Azienda Zero, Padova, Italy, 2 National Centre for Disease Prevention and Health Promotion, National Institute of Health, Rome, Italy, 3 Immunology and Diagnostic Molecular Oncology Unit, Veneto Institute of Oncology IOV-IRCCS, Padua, Italy, 4 Department of Cardiological, Thoracic and Vascular Sciences and Public Health, University of Padua, Padua, Italy

* silvia.gori@iov.veneto.it

## Abstract

### Objectives

The study aims to analyze care and cost patterns in the Veneto region for cancer patients in the final phase of their disease.

### Methods

The study cohort includes adult patients alive on 1.1.2018, which have been diagnosed with cancer at the age 15+ in a 28-years period, spanning from 1990 to 2017, and died within 12 months from prevalence date. The end-of-life (EOL) prevalent cases included in the study cohort are identified by the Veneto Tumor Registry. Primary tumors considered in the analysis are colon, rectum, skin melanoma, female breast, thyroid. Patient monthly average EOL costs, patient annual average EOL costs and total regional costs have been estimated separately for each cancer site/sex/age group/health care service database.

### Results

The estimated EOL total cost, for the five analyzed cancer types in the Veneto Region, is approximately 53 million euros. Costs associated with cancer treatment tend to increase in the final phase of the disease. In general, Hospital Admission is the main driver cost in all types of cancer considered, with the only exception in melanoma, where Hospital Drugs features the highest contribution. Moreover, costs differ significantly depending on the type of cancer, gender, and age, revealing highest average costs associated with younger patients.

### Conclusions

This study provides a comprehensive analysis of healthcare budget distribution in the last year of life, enabling healthcare planners to predict cancer burden in the future. This model can be applied to other Italian regions and countries with different health-care systems.

**Data availability statement:** All relevant data are within the manuscript and its Supporting information files.

**Funding:** SF - This study was funded by the Italian Ministry of Health (Ricerca finalizzata 2018, Grant Number RF-2018-12365530). The funder had no role in the design and conduct of the study; collection, management, analysis, and interpretation of the data; preparation, review, or approval of the manuscript; and decision to submit the manuscript for publication.

**Competing interests:** The authors have declared that no competing interests exist.

## Introduction

Cancer is a significant health issue, costing 199 billion in Europe in 2018, with 103 billion attributed to direct costs [1]. In Italy, the share of public health expenditure due to cancer is 6.7%, consistent with the European average. The total cost of cancer and its development are influenced by disease burden and technological advancements. Population aging is a major factor in cancer burden growth. In Europe, 5% of individuals live after a cancer diagnosis, with Italy having 3.6 million people in 2020 [2]. Cancer prevalence is expected to grow by 3% annually [3], with 4 million prevalent cases expected in Italy by 2030 [4].

The direct health care cancer cost curve follows a U-shape distribution, starting high at diagnosis, plateauing during relapse monitoring, and then increasing again [5–7]. The largest monthly expenditure for cancer patients is in the final few months of life [8], with up to four times higher per capita healthcare costs for those at the end of life (EOL) compared to age-matched individuals who are not at the EOL [9]. In United States, Mariotto et al. [10] found that costs in the EOL cancer phase were substantially higher than in the initial phase, suggesting a "J-shaped" curve. According to the authors, this J-shaped curve reflects a greater treatment intensity at the EOL, especially for patients with distant disease and poor prognosis cancers. Indeed, costs in the EOL phase vary according to the cancer site, thus reflecting differences in prognosis and treatments. EOL patients represent a small proportion of the total number of cancer patients diagnosed, ranging from about 3% for tumors with a better prognosis (women with breast cancer) to about 21% for tumors with the worst prognosis (men with lung cancer) [11]. In Italy, it has been estimated that these patients account for a significant proportion of total direct healthcare expenditure in terms of inpatient and outpatient services and prescription drugs, ranging from 20% for rectal cancer [7], to 14% for breast cancer [6].

Over the past 20 years, there has been a rise in observational studies examining resource utilization and associated costs at EOL in adult cancer patients [12]. Individuals' health care needs increase as they approach death, with over a quarter of all acute healthcare costs incurred in the last year of life. However, the measurement of growth of health care expenditures at the EOL remains poorly documented in the literature [13]. Understanding these drivers can help forecast future expenditures and inform policies to control cancer burden.

The majority of costs for cancer patients at EOL are derived from acute inpatient care [14], drug prescription expenditures, and outpatient care. However, patients increasingly use facilities like Hospice, Integrated Home Care, and Local Residential Care, which provide palliative care services. These healthcare expenses are rarely considered in estimating direct costs due to incomplete or difficult data access.

The aim of this study is to estimate and describe patterns of care and cost profiles associated with the health care of cancer patients in the final phase of the disease in the Veneto region, using data collected in the second edition of the Epicost-study (Epicost-2 database).

## Methods

### Context

The structure of the Italian National Health System (NHS) relies primarily on general taxation and operates as a regionally organized public service. Its principles are rooted in universal coverage, unrestricted access, freedom to choose, pluralism in service provision, and fairness [15]. Regional authorities plan and organize healthcare facilities and activities in accordance with a national health plan designed to assure an equitable provision of comprehensive care, called essential levels of assistance. Veneto is a region in northeastern Italy. It has a resident population of 4.8 million with a mean age of 46.6 years.

## Study design

The Epicost-study, using a multi-disciplinary methodology that involves epidemiological, health economic and statistical components, aims to measure in Italy direct medical costs sustained by the Italian NHS for services provided to cancer patients, along their disease pathway, using microdata integrated from multiple population-based data sources. The study is a retrospective, population-based study identifying prevalent cases for different cancer types (accessed date on 25.10.2023). All data were fully anonymized. More details about features and methods used in the Epicost-study are provided in Francisci et al. [16].

## Inclusion criteria

The cost analysis includes patients alive on 1.1.2018 (prevalence date), who were diagnosed with cancer at the age 15 years or older during the 28-year period from 1990 to 2017, and who died of malignancy within 12 months of the prevalence date, i.e., during the period from January 1st 2018 to December 31st 2018. The EOL prevalent cases are identified by the Veneto Tumor Registry (VTR) and refer to about half of the regional population. In order to extend estimates of the EOL costs to the whole region, prevalent cases of the whole region were obtained using the COMPREV software.

Index cancers considered in the analysis are the following: colon (ICD-X C18), rectum (ICD-X C19-C20), skin melanoma (ICD-X C43), female breast (ICD-X C50), thyroid (ICD-X C73). Individuals with multiple tumor diagnoses within 5 years from the cancer considered are excluded, as the attribution of costs to the index tumor would be uncertain. Information about multiple tumors is provided by the VTR.

## Materials

Each EOL prevalent case is linked at individual level with health care services databases in a twelve-month period backwards from death to estimate costs related to all procedures, interventions, devices and drugs administered to the patient in the EOL. Cost is defined here as the reimbursement claimed to the regional health authority for the health care service provided.

Health care service claims are from the following health care services databases: Hospital Admission (HA), Outpatient Services (OPS), Drug Prescriptions (DP), Hospital Drugs (HD), Hospice (HSP), Integrated Home Care (IHC), Emergency Room (ER), Medical Devices (MD) and Local Residential Care (LRC). The first four listed databases are called main and the last five are called additional.

Diagnostic procedures and interventions in HA, OPS, HSP, ER databases are classified according to the International Classification of Diseases (ICD9-CM); drugs in DP and HD databases are classified according to the Anatomical Therapeutic Chemical Classification System (ATC); diseases in the IHC and LRC databases are classified according the International Classification of Primary Care (ICPC) code; the type of device in the MD database is classified according to the ISO code.

The HA database contains demographic information (date of birth, sex, place of residence), clinical information (main and secondary diagnosis, main and secondary interventions) and total claim in Euros based on the DRG system.

The OPS database provides information on outpatient services (for example, diagnostic tests and ambulatory interventions such as chemotherapy and radiotherapy) and total claim in Euros.

The DP database includes information on drugs delivered by pharmacy; the HD database contains data on high cost drugs administered to a patient during hospitalization or in outpatient setting. Both DP and HD databases contain drug ATC code, date of prescription and total claim in Euros.

The HSP database contains records referring to hospice staying. It includes date of admission, date of discharge, the main diagnosis, the daily rate. The cost of hospice is calculated by multiplying the daily rate by the days of staying.

The IHC source contains information on home visits conducted by doctors, nurses or other health professionals. Each record includes the date of access, the type of health professional involved, the codes of the main and secondary diseases, the number of accesses per day. The cost of each record depends on the costs of different health care personnel and the number of accesses per day.

The ER source contains records each one referring to a single undergone intervention during the stay in the emergency room. It includes date of admission, date of discharge, main and secondary diagnosis, code of undergone intervention, total claim in Euros.

The MD source collects information regarding medical devices provided to patients. Each record contains the delivery date, the type of device, the amount delivered, the cost of the single device. The cost of each record is calculated by multiplying the cost of the single device by the amount delivered.

The LRC source contains information on residency staying. The stay can be suspended for a period, most of all due to hospital admission. Each record includes date of residency admission, date of residency discharge, if applicable date of suspension and date of readmission, the main and secondary diseases, the daily rate. Each patient cost is calculated by multiplying the daily rate by the number of days of staying, and half daily rate by the number of days of suspension.

In order to identify EOL costs related to cancer in the baseline databases, lists of codes, specific for each cancer site and classification system (ICD9-CM for diagnosis and procedures, ATC for drugs and ICPC for diseases) are used. These lists have been obtained by using either the attribution method as is the case for breast female, colon and rectum [17] or an indirect approach similar to that applied in the US for Medicare-SEER cost analysis [10] as is the case for thyroid and melanoma.

As concerning the additional databases, all costs related to services provided to EOL patients were included in the analysis, except for IHC and LRC, where the cost is included only if the main and/or secondary diseases are related to a cancer diagnosis.

## Statistical analysis

The estimates of the complete EOL prevalence cases in the registration area are obtained by applying the methodology implemented in the phase-of-care session of the COMPREV software [11]. According to the requirements of the software, two Limited Duration Prevalence (LDP) matrices are used as input data: one refers to 01/01/2018 and the other to 01/01/2019. The complete EOL prevalence in the whole region was then obtained using the proportions of prevalence estimates (age-, sex-, and cancer type-specific) obtained for the half-regional area covered by cancer registration, multiplied by the regional population by sex and age.

Estimates refer to the year 2018 and are provided for 5 cancer sites representing overall the 34.2% of the total cancer incidence in 2017 and 48.8% of the total cancer prevalence at January 1st 2018. EOL prevalent cases are subdivided according to the cause of death into: EOL_C, including those patients dying for any malignant cancer (C00-C97); EOL_NC, including those patients dying for any cause other than malignant cancer.

The cost indicators: $C_k^{EOL}$, $C^{EOL}$ and Tot regional costs have been estimated separately for each cancer site/sex/age group/health care service database [16].

In detail:

- Patient monthly average EOL costs $C_k^{EOL}$ are obtained by dividing costs of all EOL patients in month k by the corresponding person-months; the sequence of the twelve patient monthly average costs in the final phase of care constitutes the EOL cost profile.

- Patient average costs for the last year of life $C^{EOL}$ are computed as follows: costs of all EOL patients over twelve months are divided by the corresponding person-months and this ratio is then multiplied by the 12 months of the final phase.

- Total regional costs related to cancer care in the last year of life, obtained by applying the $C^{EOL}$ by cancer site/sex/age group/health care service database to the estimates of complete EOL prevalence for the Veneto region in 01/01/2018.

The average monthly costs were described in terms of summary statistics (mean, median, minimum, maximum and standard deviation) overall and by sex/age. Costs were compared by sex using the parametric t-test or the analogous nonparametric Mann-Whitney test, and nonparametric Kruskal-Wallis test by age group, due to the non-Normal distribution. Comparisons among age groups were performed using Dunn's Test with Bonferroni adjustment. The average annual costs of the study cohort have been compared through the nonparametric Mann-Whitney test (due to the non-Normal distribution) with the same costs of VTR population diagnosed with one of the five cancers in the years 2014–2017, period in which VTR collected information about all the regional population.

Statistical significance was considered for p values less than 0.05. Statistical analyses were performed using the R software, release 4.2.1 (R Core Team, 2022) [18] and SAS Enterprise Guide software, release 6.1 (SAS Institute, Cary NC).

## Ethics statement

Ethical review and approval was not required for the study on human participants in accordance with the local legislation and institutional requirements. Written informed consent for participation was not required for this study in accordance with the national legislation and the institutional requirements.

## Results

Overall, the prevalence cohort includes 3,157 subjects. The majority are female (67.2% vs. 32.8%) and aged over 70 years (69.6%) (Table 1).

Analyzing the distribution by gender (excluding breast cancer), only thyroid cancer shows a higher prevalence in females than males. According to the distribution by age, a low prevalence percentage is observed in the younger age group, never exceeding 8%, while the highest prevalence is in the 70+ age group. This trend is particularly pronounced in the colon cancer (1.7% and 77.1% of prevalent cases, respectively).

For the cancer types analyzed, no statistically significant difference was found between average annual costs of the study cohort and the VTR population diagnosed with a cancer in study between 2014–2017 (p-value equal to 0.94).

Focusing on average cost per patient, as expected, it is higher in younger patients, decreasing with age. Melanoma has the highest total average cost in EOL phase per patient, followed by rectum and colon, and it is more than twice of breast and thyroid cancers.

Fig 1 shows the contribution of baseline and additional health care services databases to the average annual cost by cancer type. The primary contribution comes from baseline databases (range 66.7%–90.9%), but additional costs represent a significant share ranging from 9.1% for melanoma to 33.3% for thyroid.

Stratifying the result by health care service databases, it can be noticed that HA is the main cost driver in all the considered types of cancer; with the only exception in melanoma, where HD features the highest percentage (Fig 2).

**Table 1. EOL prevalent cases by cancer type, sex and age group. In brackets the average cost.**

| Cancer type | | Gender | | Age group | | | Total |
|---|---|---|---|---|---|---|---|
| | | Male | Female | 15–49 y | 50–69 y | 70+ y | |
| **Colon** | N (%) | 567 (56.6%) | 434 (43.4%) | 17 (1.7%) | 212 (21.2%) | 772 (77.1%) | 1,001 (31.7%) |
| | Average cost | [19,207.45] | [17,990.98] | [44,557.67] | [25,406.42] | [15,589.44] | [18,609.78] |
| **Breast** | N (%) | 0 (0%) | 1,356 (100%) | 98 (7.2%) | 372 (27.4%) | 886 (65.3%) | 1,356 (43.0%) |
| | Average cost | [0] | [12,906.18] | [25,750.04] | [15,849.11] | [10,324.06] | [12,906.18] |
| **Melanoma** | N (%) | 176 (64.0%) | 99 (36.0%) | 17 (6.2%) | 67 (24.4%) | 191 (69.4%) | 275 (8.7%) |
| | Average cost | [28,621.87] | [25,612.28] | [51,057.85] | [35,102.55] | [16,906.24] | [27,301.88] |
| **Rectum** | N (%) | 265 (59.4%) | 181 (40.6%) | 11 (2.5%) | 134 (30%) | 301 (67.5%) | 446 (14.1%) |
| | Average cost | [19,847.51] | [19,551.18] | [27,083.82] | [26,599.54] | [16,138.32] | [19,721.85] |
| **Thyroid** | N (%) | 28 (35.4%) | 51 (64.6%) | 3 (3.8%) | 28 (35.4%) | 48 (60.8%) | 79 (2.5%) |
| | Average cost | [5,069.92] | [13,713.47] | [0] | [14,836.00] | [9,372.99] | [11,053.91] |
| **Total prevalent cases** | | 1,036 (32.8%) | 2,121 (67.2%) | 146 (4.6%) | 813 (25.8%) | 2,198 (69.6%) | 3,157 (100%) |

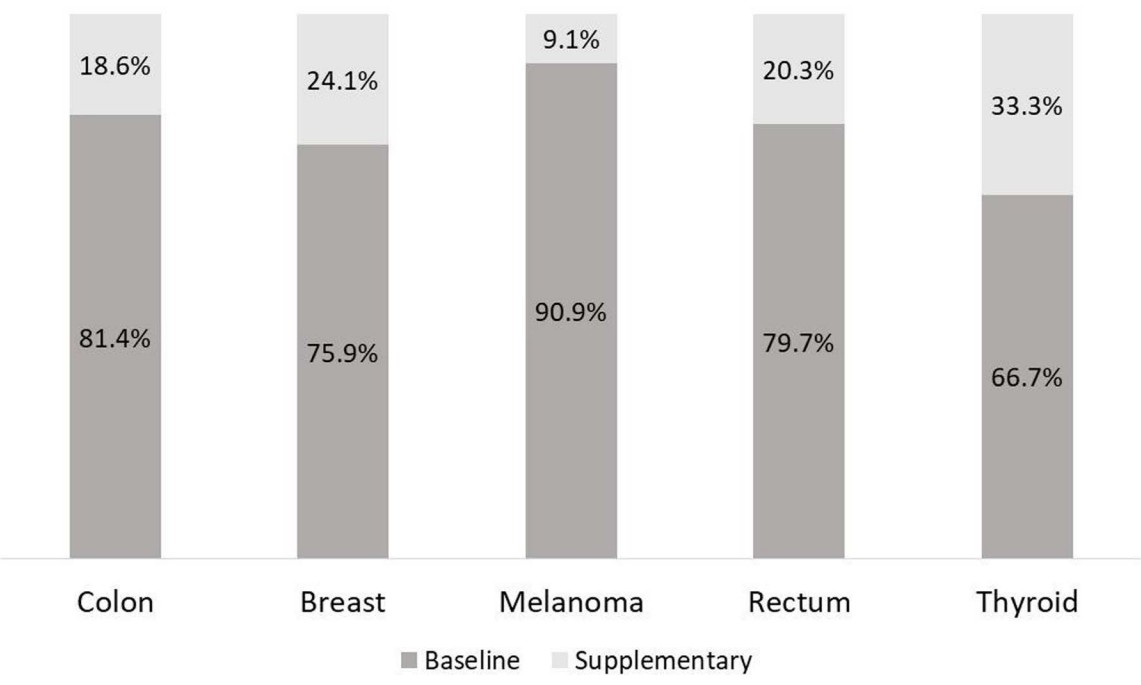

**Fig 1. Contribution of baseline and additional health care services databases to the average annual cost by cancer type.**

HA costs are followed by HD costs, the only exception being in thyroid cancer where the second most expensive service is HSP. The OPS database contributes above 12% in all cancer types, except for melanoma where it is below 3%.

Between additional sources the highest contribution is provided by HSP in almost all cases.

The average and total annual costs are reported in S1 Table of supporting information, stratified by cancer type and health care service database.

Fig 3 shows the distribution of total annual cost by cancer type (histogram) and the total number of prevalent cases (line).

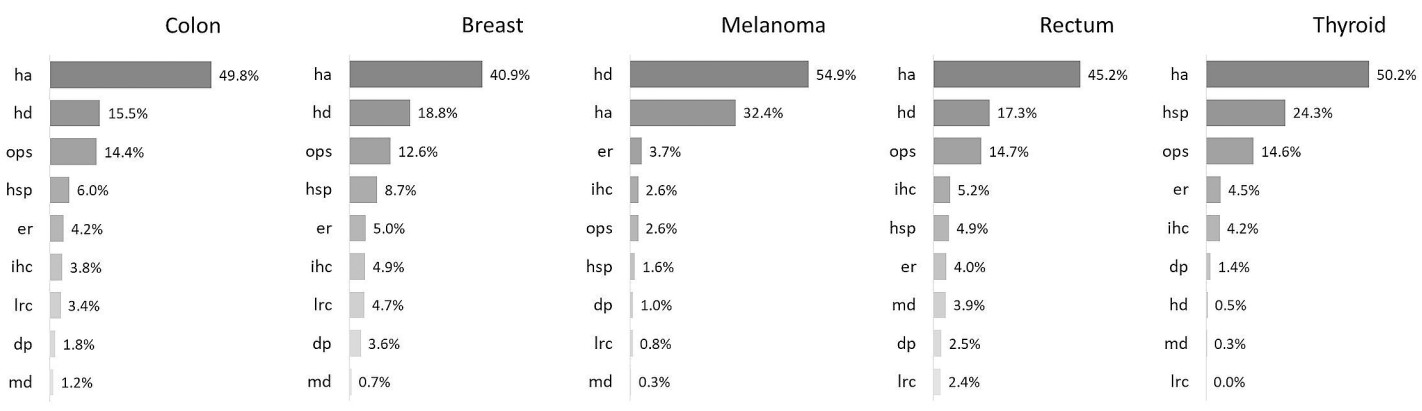

**Fig 2. Share of health care service databases contributing to the total cost by cancer type.**

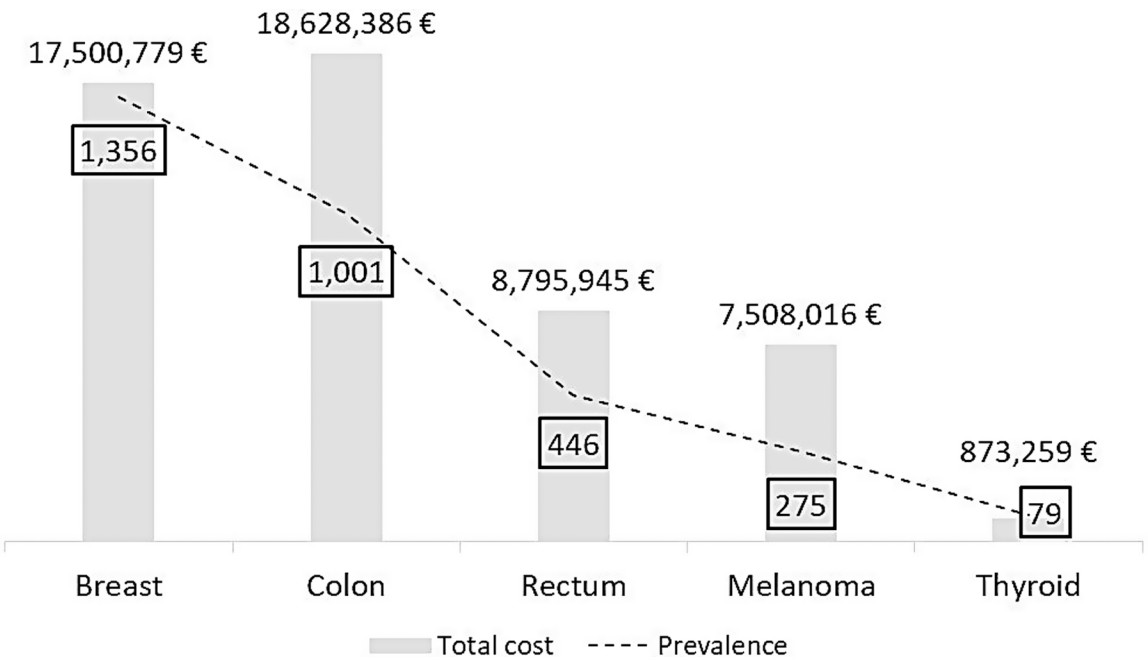

**Fig 3. Distribution of total annual cost (in euro) and prevalent cases by cancer type.**

In general, it is evident that a higher number of prevalent cases corresponds to a higher total annual cost. However, despite having fewer prevalent cases compared to breast cancer (1,001 vs. 1,356), colon cancer leads to a higher total annual cost (18,628,386 euros vs. 17,500,779 euros).

Melanoma has just over half the prevalence cases compared to rectal cancer, but its total annual cost is only 15% lower than rectal cancer cost.

The estimated total cost in the final phase, for the five analyzed cancer types is approximately 53 million euros.

The average monthly costs were analyzed by stratifying both by sex and by age group. Table 2 shows statistically significant decreasing trend in costs as age increases (p < 0.001). As expected, no gender differences in expenditures due to cancer diagnosis and treatment are present.

**Table 2. Descriptive statistics of the average monthly cost in the final phase by sex and age group.**

|  | Mean | Median | Min - Max | Standard deviation | P-value |
|---|---|---|---|---|---|
| **Male** | 166.70 | 30.31 | 0.00 - 2,615.29 | 358.46 | 0.0398[a] |
| **Female** | 176.23 | 43.83 | 0.00 - 3,237.44 | 371.79 |  |
| **15–49 y** | 273.56 | 0.00 | 0.00 - 5,776.14 | 729.85 | <0.001[b] |
| **50–69 y** | 217.02 | 30.69 | 0.00 - 3,538.00 | 460.17 |  |
| **70+ y** | 124.86 | 40.13 | 0.00 - 2,201.09 | 261.82 |  |

[a]Nonparametric Mann-Whitney U test. Breast cancer excluded.

[b]Nonparametric Kruskal-Wallis H test, pairwise comparison using Dunn's Test with Bonferroni adjustment. Statistical significant difference between 15–49 y and 50–69 y (p-value <0.001), and between 15–49 y and 70+ y (p-value <0.001).

In Fig 4, the trends of average monthly costs in the last 12 months of life are reported for each cancer type, considering only health care services databases with at least a 5% contribution to the average annual cost as reported in Fig 2, (HA, dashed line, was placed it on a secondary axis).

Regarding colon cancer (Fig 4a), it is notable that the average monthly cost of HA remains stable between 500 and 1,000 euros up to 2 months before death, then increases to over 2,000 euros in the last month of life. Showing a similar trend, HSP has a constant, almost negligible, average monthly cost until the 7th month, after which it increases until the last month of life, reaching approximately 600 euros. OPS and HD, instead, show a reverse trend compared to HA and HSP; they have a steady average monthly cost around 200–300 euros, followed by a decrease in the last 3 months of life.

OPS and HD also show a statistically significant difference in average monthly costs by sex (p = 0.017 and p = 0.033, respectively), as well as among age groups (p = 0.007 and p = 0.001, respectively) (S2 and S3 Tables).

Regarding breast cancer (Fig 4b), it is observed that the trends of average monthly costs for HA and HSP are very similar, increasing gradually up to 2 months before death and then rising sharply in the last month of life (2,000 euros and 500 euros, respectively). ER maintains a constant trend at about 50 euros up to 5 months before death, then constantly increases in the last four months, reaching almost 150 euros in the last month. OPS follows a constant trend up to 4 months before death, then increases in the 3rd month, and finally decreases in the last two months of life. OPS and HD, along with DP, showed a statistically significant difference in average monthly costs by age group (p < 0.001) (S3 Table).

Melanoma (Fig 4c) has only two databases contributing at least 5% to the average annual costs: HD and HA. These databases show a similar pattern with the only difference being their opposite trends in the last three months of life: while HA increases, HD decreases. Melanoma shows a statistically significant difference in average monthly costs among age groups overall (p = 0.030). In addition, it has been found a statistically significant difference in HD, DP, and OPS among age groups (p < 0.001, p = 0.004, p = 0.020 respectively) (S3 Table).

Regarding rectal cancer (Fig 4d), it is observed that the average monthly cost of HA ranges from 400 to 600 euros in the first eight months, then increases in the last four months, reaching almost 2,000 euros in the last month of life. Similarly, IHC increases gradually during the period, reaching about 300 euros in the last month. OPS and HD, on the contrary, exhibit a reverse trend: after a constant pattern, they start decreasing in the last three months. As well as melanoma, in rectal cancer there is a statistically significant difference in average monthly costs among age groups considering all health care service databases (p < 0.001), and also specifically for HD, DP and OPS (p = 0.013, p = 0.011, p < 0.001 respectively) (S3 Table).

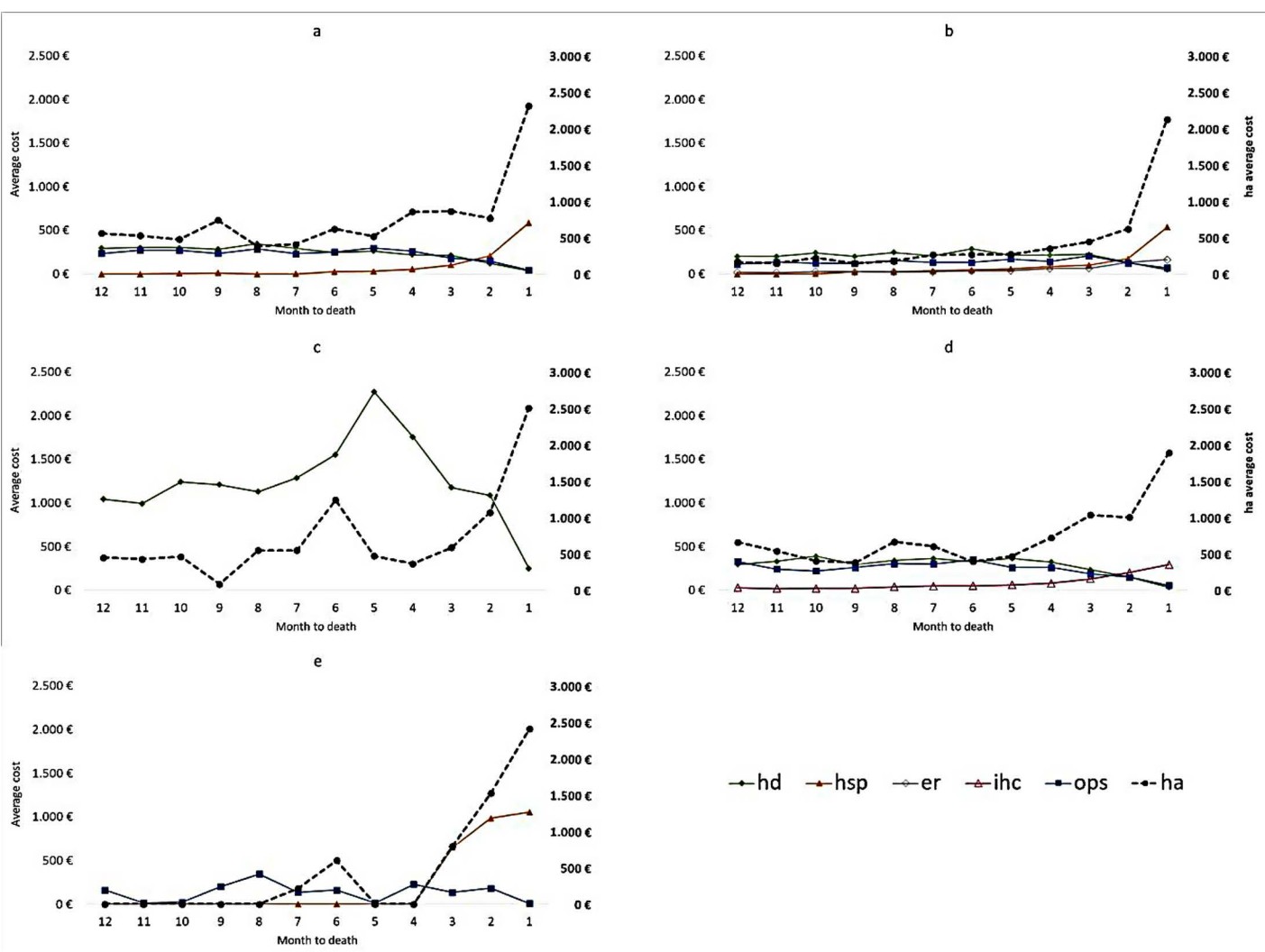

**Fig 4. Trends of average monthly costs in the last 12 months of life for the most expensive services in each type of cancer. (a) colon, (b) breast, (c) melanoma, (d) rectum, (e) thyroid.**

Fig 4e describes the cost trends for thyroid cancer. The pattern is similar to the other cancer types, with a consistent increase in the last three months of life for both HA and HSP, reaching up to about 2,500 euros and 1,000 euros, respectively, and a decrease in the last months for OPS. The low number of prevalent cases (only 13 patients for the cost calculation) doesn't allow a full view of the trend, having no cost in many months. For this reason, for thyroid cancer, statistically significant differences by gender and age group were not considered.

## Discussion

This paper used routine healthcare data linked with cancer registry data to provide from a public health perspective an estimate of health service use and costs of cancer patients at the EOL.

The study found that resource utilization increases sharply as death approaches, particularly in the last 3 months of life, consistent with previous literature [10,19]. Moreover, resource use decreases with increasing age [20], possibly due to different clinical approaches.

Young patients better tolerate aggressive treatments and have a longer survival chance compared to elderly patients, who are more exposed to comorbidities. There are also disparities in the utilization rate of innovative drugs according to age [13]. Additionally, the chance of receiving intensive or life-sustaining treatments decreases with age [9], consistent with cost estimates of rectal and breast female cancer patients [6,7]. However, proximity to death affects healthcare expenditure even more than age [12].

Hospital admission is the primary driver of resource use and expenditures for all cancers, except skin melanoma. This is consistent with international literature, where hospital admissions account for a greater proportion of costs than other services for cancer patients at the EOL [9,10]. Hospice care, often underutilized, is linked to reduced resource utilization and aggressive EOL treatment compared to hospital-based approaches. This raises questions about the quality and appropriateness of cancer care. Palliative care is recommended for EOL patients to improve the standards of care and reduce health costs [21]. Recent studies suggest that hospital-centered systems should move towards decentralized systems, such as community-centered and home-centered, for both non-emergency cases and terminally ill patients. This is important for saving public money, avoiding cost increases, and improving healthcare services and patient end life quality [22].

The study reveals that despite similar care and cost profiles across various cancer types, there are significant differences in the total resources used to care for cancer patients, including those at the EOL, due to variations in patient costs and the number of patients.

Melanoma is the most expensive cancer due to advanced therapies, such as targeted therapy or immunotherapy, but it contributes to 14% of EOL care expenditures for all five cancer types. Breast and colon cancers, with costs half and two-thirds of melanoma patients, contribute 67% of total EOL expenditures. Thyroid cancer, with a cost comparable to breast cancer, only contributes 1%. These findings align with previous research showing that cancer type is associated with resource use [10,19]. The low number of prevalent cases contributes to the overall EOL care expenditures.

The main strength of the Epicost-study, in particular with the second release of the study, was that it allowed the measurement of all costs attributable to healthcare services provided to cancer patients in the last year of life, thanks to the integration of a wide range of healthcare databases, including hospice, integrated home care and local residential care.

Despite their relevance in terms of share of EOL costs (varying from 9.1% for skin melanoma to 33.3% for thyroid cancer), these databases are less commonly accounted, due to issues related to accessibility, completeness and standardizations of the information collected.

Moreover, this real-world study provides reliable estimates of all costs directly attributable to healthcare services used by patients in their last year of life, using as much data sources as available at the time of data collection. However, some pitfalls are to mention. First of all, there are few additional health care service databases not included in the present analysis, since their collection started in 2017 after the establishment of the Epicost-2 study protocol: community hospitals (CH) and territorial rehabilitation units (TRU). From an a-posteriori sensitive analysis resulted that costs traced from these flows represent, for the five cancers considered, less than 3% of the costs due to the additional databases. Secondarily, costs refer to the 2018 prevalence cohort: it does not allow incorporating rapidly changing treatment and cost patterns [20,23]; however, it does not require a temporal price, as all costs refer to the common reference year 2018 [16].

Thirdly, other limitations are related to the data from the health care services databases, in fact data on claims are collected for administrative rather than for research (epidemiological) purposes. This might affect the quality and the completeness of the information on health care service expenditures. Finally, in order to identify prevalent cases in the EOL, information on the cause of death is needed. The coding of death certificates might be affected by inaccuracy

and inconsistency issues, thus limiting the comparability of cause of death information across different areas/populations.

## Conclusions

In the context of population ageing and increasing costs of health care particularly for EOL patients, there is increasing debate about the need for evidence supporting that treatment intensity reflects patient goals of care, in order to ensure value for patients while preserving and enhancing quality and sustaining innovation [24].

The Epicost-study contributes to this debate, by allowing a comprehensive description about the amount and the distribution of health care budget in the last year of life according to different health care service components and patients' characteristics, therefore it can be used by health care planners to make predictions of cancer burden into the near future according to specific interventions and corresponding scenarios.

The model of analysis proposed here is replicable to other Italian regions and possibly to other countries with different health care systems, provided that individual health care information on services and corresponding claims are available.

## Supporting information

**S1 Table. Annual average costs, complete prevalent cases and total costs in the final phase by cancer type and health care services database.**
(DOCX)

**S2 Table. Descriptive statistics of the average monthly cost in the final phase by sex, cancer type and health care services database.**
(DOCX)

**S3 Table. Descriptive statistics of the average monthly cost in the final phase by age group, cancer type and health care services database.**
(DOCX)

## Author contributions

**Conceptualization:** Stefano Guzzinati, Tania Lopez, Manuel Zorzi, Silvia Francisci.

**Data curation:** Stefano Guzzinati, Daniela Pierannunzio, Andrea Tavilla.

**Formal analysis:** Stefano Guzzinati, Alessandra Andreotti, Silvia Gori, Alberto Gagliani, Silvia Francisci.

**Funding acquisition:** Silvia Francisci.

**Investigation:** Stefano Guzzinati, Alessandra Andreotti, Tania Lopez, Silvia Gori, Alberto Gagliani, Silvia Francisci.

**Methodology:** Stefano Guzzinati, Alessandra Andreotti, Tania Lopez, Silvia Gori, Alberto Gagliani, Silvia Francisci.

**Project administration:** Stefano Guzzinati, Tania Lopez, Silvia Francisci.

**Resources:** Stefano Guzzinati, Silvia Francisci.

**Software:** Stefano Guzzinati, Alessandra Andreotti, Silvia Gori, Alberto Gagliani, Daniela Pierannunzio, Andrea Tavilla, Silvia Francisci.

**Supervision:** Stefano Guzzinati, Silvia Francisci.

**Validation:** Stefano Guzzinati, Alessandra Andreotti, Tania Lopez, Silvia Gori, Alberto Gagliani, Silvia Francisci.

**Visualization:** Stefano Guzzinati, Alessandra Andreotti, Tania Lopez, Silvia Gori, Alberto Gagliani, Silvia Francisci.

**Writing – original draft:** Stefano Guzzinati, Alessandra Andreotti, Tania Lopez, Silvia Gori, Alberto Gagliani, Andrea Tavilla, Alessandra Buja, Silvia Francisci.

**Writing – review & editing:** Stefano Guzzinati, Alessandra Andreotti, Tania Lopez, Silvia Gori, Alberto Gagliani, Sandra Mallone, Daniela Pierannunzio, Andrea Tavilla, Alessandra Buja, Manuel Zorzi, Silvia Francisci.

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
