## [Decision Letter · Decision Letter 0]

23 Jan 2025

Cost profiles of cancer patients at the end of life: estimates from the EPICOST-study

PONE-D-24-52487

Dear Dr. Gori,

We’re pleased to inform you that your manuscript has been judged scientifically suitable for publication and has been be formally accepted for publication

Kind regards,

Monia Marchetti

Academic Editor

PLOS ONE

Journal Requirements:

Additional Editor Comments (optional):

Dear Author

we are pleased to inform you that you manuscript has been successfully accepted for publication.

Academic Editor PLOSONE

Reviewers' comments:

Reviewer's Responses to Questions

**Comments to the Author**

1. Is the manuscript technically sound, and do the data support the conclusions?

Reviewer #1: Yes

2. Has the statistical analysis been performed appropriately and rigorously? 

Reviewer #1: Yes

3. Have the authors made all data underlying the findings in their manuscript fully available?

Reviewer #1: Yes

4. Is the manuscript presented in an intelligible fashion and written in standard English?

Reviewer #1: Yes

5. Review Comments to the Author

Reviewer #1: The manuscript is coherent in all its parts, it is robust in the methodology and rich for the data sources used. The time frame considered to measure costs in EOL is sufficient. Literature citations are appropriate

6. PLOS authors have the option to publish the peer review history of their article (what does this mean? ). If published, this will include your full peer review and any attached files.

**Do you want your identity to be public for this peer review?** For information about this choice, including consent withdrawal, please see our Privacy Policy .

Reviewer #1: No

---

## [Editor Report · Acceptance letter]

PONE-D-24-52487

PLOS ONE

Dear Dr. Gori,

I'm pleased to inform you that your manuscript has been deemed suitable for publication in PLOS ONE. Congratulations! Your manuscript is now being handed over to our production team.

Kind regards,

on behalf of

Dr. Monia Marchetti

Academic Editor

PLOS ONE